# The High Capacity of Brazilian *Aedes aegypti* Populations to Transmit a Locally Circulating Lineage of Chikungunya Virus

**DOI:** 10.3390/v16040575

**Published:** 2024-04-09

**Authors:** Amanda de Freitas, Fernanda Rezende, Silvana de Mendonça, Lívia Baldon, Emanuel Silva, Flávia Ferreira, João Almeida, Siad Amadou, Bruno Marçal, Sara Comini, Marcele Rocha, Hegger Fritsch, Ellen Santos, Thiago Leite, Marta Giovanetti, Luiz Carlos Junior Alcantara, Luciano Moreira, Alvaro Ferreira

**Affiliations:** 1Mosquitos Vetores: Endossimbiontes e Interação Patógeno-Vetor, Instituto René Rachou-Fiocruz, Belo Horizonte 30190-002, Brazil; acfreitas@aluno.fiocruz.br (A.d.F.); fernanda.rezende@fiocruz.br (F.R.); smendonca@aluno.fiocruz.br (S.d.M.); livia.baldon@aluno.fiocruz.br (L.B.); bmarcal@aluno.fiocruz.br (B.M.); scomini@aluno.fiocruz.br (S.C.); marcelebio@yahoo.com.br (M.R.); hegger.fritsch@gmail.com (H.F.); giovanetti.marta@gmail.com (M.G.); luiz.alcantara@fiocruz.br (L.C.J.A.); luciano.andrade@fiocruz.br (L.M.); 2Departamento de Bioquímica e Imunologia, Instituto de Ciências Biológicas, Universidade Federal de Minas Gerais, 6627-Pampulha, Belo Horizonte 31270-901, Brazil; emanuelegsilva@gmail.com (E.S.); fvianaferreira@gmail.com (F.F.); joaopaulobio@ufmg.br (J.A.); gbadeguetchin@gmail.com (S.A.); ellencaroline24@gmail.com (E.S.); thjfl21@gmail.com (T.L.); 3Department of Sciences and Technologies for Sustainable Development and One Health, University of Campus Bio-Medico, 00128 Rome, Italy

**Keywords:** chikungunya virus, *Aedes aegypti*, vector competence, arbovirus

## Abstract

The incidence of chikungunya has dramatically surged worldwide in recent decades, imposing an expanding burden on public health. In recent years, South America, particularly Brazil, has experienced outbreaks that have ravaged populations following the rapid dissemination of the chikungunya virus (CHIKV), which was first detected in 2014. The primary vector for CHIKV transmission is the urban mosquito species *Aedes aegypti*, which is highly prevalent throughout Brazil. However, the impact of the locally circulating CHIKV genotypes and specific combinations of local mosquito populations on vector competence remains unexplored. Here, we experimentally analyzed and compared the infectivity and transmissibility of the CHIKV-ECSA lineage recently isolated in Brazil among four *Ae. aegypti* populations collected from different regions of the country. When exposed to CHIKV-infected AG129 mice for blood feeding, all the mosquito populations displayed high infection rates and dissemination efficiency. Furthermore, we observed that all the populations were highly efficient in transmitting CHIKV to a vertebrate host (naïve AG129 mice) as early as eight days post-infection. These results demonstrate the high capacity of Brazilian *Ae. aegypti* populations to transmit the locally circulating CHIKV-ECSA lineage. This observation could help to explain the high prevalence of the CHIKV-ECSA lineage over the Asian lineage, which was also detected in Brazil in 2014. However, further studies comparing both lineages are necessary to gain a better understanding of the vector’s importance in the epidemiology of CHIKV in the Americas.

## 1. Introduction

Chikungunya virus (CHIKV) is a major threat to global public health, and over the last three decades, it has been responsible for epidemics that are increasing in frequency and geographic scale [1,2,3]. Currently, CHIKV circulates throughout the tropical and subtropical regions of Africa, South Asia, and South America [4,5,6]. CHIKV is an arthropod-borne virus (arbovirus) that is transmitted mainly by *Aedes* mosquito species, and the infection commonly results in a febrile illness called chikungunya fever (CHIKF), which is usually self-limiting [1,3,7]. The main symptom of CHIKF is chronic and severe joint pain, which can be accompanied by an itchy maculo-papular skin rash and can sometimes lead to prolonged periods of functional disability. Severe complications, such as encephalitis, and fatal outcomes may occur in individuals with comorbidities [3].

CHIKV is an alphavirus (family *Togaviridae*) that has a positive sense RNA genome that encodes four non-structural proteins (nsP1–4) and five structural proteins (C, E3, E2, 6K, and E1) expressed from a subgenomic RNA [8]. Historically, CHIKV originated in sub-Saharan Africa, where a transition occurred from an ancestral enzootic sylvatic cycle involving arboreal mosquitoes and nonhuman primates to an urban cycle where peridomestic mosquitoes transmit the virus among humans [7]. According to phylogenetic studies, CHIKV first emerged in the urban cycle that encompasses humans in the eastern region of Africa before spreading to other parts of the world [4,6]. The virus is currently classified into three genotypes: West African, East-Central-South-African (ECSA), and Asian. More recently, the ECSA genotype has given rise to the Indian Ocean lineage (IOL), which has spread globally and caused numerous outbreaks since 2005.

The distribution of CHIKV has rapidly expanded since its spread from Africa. The virus is currently being transmitted locally in more than 100 countries across various regions, including the Americas, the Caribbean, the Western Pacific, Southern Europe, Southeast Asia, and Oceania [9,10,11,12,13,14,15,16,17,18,19]. Due to its geographical spread and ability to cause incapacitating disease, CHIKV was included by the World Health Organization (WHO) in the Priority Blueprint under the Global Arbovirus Initiative (GAI), which was launched in 2022 to increase global awareness of the potential risk of epidemics [20]. Indeed, the number of outbreaks of CHIKV has increased dramatically in recent years, especially in the South American region. Currently, Brazil has the highest number of chikungunya cases across the entire region [21,22].

There are currently two confirmed CHIKV lineages circulating in Brazil: the Asian lineage and the African ECSA lineage, both introduced in 2014 [22,23,24,25]. Nonetheless, it appears that the ECSA lineage is the primary genotype currently in circulation in the country [21,25,26,27,28]. Although the epidemiology of CHIKV depends on several complex factors, including host and viral factors, vector factors such as the intrinsic infection ability and transmission efficiency play an important role in the dynamics and establishment of different lineages. In Brazil, the main vectors associated with the transmission of these CHIKV lineages are the mosquitoes *Aedes aegypti* and *Aedes albopictus*, both of which are widely distributed across the country [7]. Previous studies have shown that these two *Aedes* species are competent vectors for both the ECSA and Asian lineages across different regions of the world [13,29,30,31,32,33,34,35,36]. However, in Brazil, the impact of locally circulating CHIKV genotypes and specific combinations of local *Aedes aegypti* populations on the vector competence has not been thoroughly examined. Here, we evaluated and compared the infectivity and transmissibility of a CHIKV-ECSA lineage recently isolated in Niterói, Brazil, among four *Ae. aegypti* populations collected from different regions of the country.

## 2. Materials and Methods

### 2.1. Mosquito Collection and Establishment of the Populations

Mosquito sampling took place in four distinct sites across Brazil, spanning four different states: Araraquara city in São Paulo state (ARA), Jaboticatubas in Minas Gerais state (JAB), Petrolina city in Pernambuco state (PET), and Porto Alegre city in Rio Grande do Sul (POA), during January and February of 2022 (Figure 1 and Table 1). These locations were chosen to offer a comprehensive depiction of the varied ecological and environmental conditions prevalent throughout Brazil’s diverse regions. The objective was to capture putative variations in mosquito populations influenced by a range of factors, including the climate, vegetation, and human activities. In terms of the climate, our sampling covered a range of types: subtropical (ARA and POA), tropical highland (JAB), and tropical semi-arid (PET). Eggs were collected using ovitraps, and they were shipped to the Laboratorio of Mosquitos Vetores (MV) at the Instituto René Rachou-Fiocruz Minas, Belo Horizonte, Brazil. All the mosquitoes were reared under insectarium-controlled conditions, 28 °C and 70–80% relative humidity, in a 12/12 h light/dark cycle. Eggs were placed in plastic trays containing two liters of filtered tap water, supplemented with fish food (Tetramin, Tetra) for hatching, and larvae were maintained at a density of 200 larvae per tray. After emerging, adults were kept in 30 cm × 30 cm × 30 cm BugDorm insect cages, where mosquitoes were fed with 10% sucrose solution ad libitum. For the establishment of each laboratory population, 100 females and 100 males were used.

### 2.2. Virus Strain, Viral Propagation and Titration

To evaluate the mosquito vector competence, we used the CHIKV-ECSA strain isolated from a human patient in Niterói, Rio de Janeiro state, Brazil, in 2018 (GenBank number—MK244647). The serum sample from the patient with CHIKV-like symptoms was sent to the Flavivirus Reference Laboratory (LABFLA—Oswaldo Cruz Institute) for molecular diagnostics. After molecular screening, nucleic acid extraction and purification were performed following the manufacturer’s recommendations and using the Magmax Pathogen RNA/DNA and KingFisher Plex Purification System (Thermo Fisher, Waltham, MA, USA) kits. The RT-qPCR for CHIKV RNA detection followed an adapted protocol from Fritsch and coauthors [22]. This CHIKV lineage was propagated in C6/36 Ae. albopictus cells. The C6/36 cells were maintained on L15 medium, supplemented with 10% FBS (fetal bovine serum) and 1× Antibiotic Antimycotic (Gibco), as described in [37]. Cells were seeded to 70% confluence and infected at a multiplicity of infection (MOI) of 0.01 and maintained for six to nine days at 28 °C. The supernatant was collected and clarified by centrifugation to generate virus stocks that were kept at −80 °C prior to use. Mock supernatants used as controls were prepared following the same procedure, without virus infection. Titration was performed in Vero cells using the plaque assay method to determine the viral titer. Titration was performed on six-well tissue culture plates. We allowed the virus to adsorb for 1 h at 37 °C, then an overlay of 2% carboxymethyl cellulose (CMC) in DMEM with 2% FBS was added. The plates were incubated at 37 °C and 5% CO_2_ for 5 days. Then, formaldehyde was added, and the cells were covered with a crystal violet stain (70% water, 30% methanol, and 0.25% crystal violet) to visualize the plaques.

### 2.3. Mice Inoculation with CHIKV

CHIKV inoculation of AG129 mice (IFN α/β/γ R^−/−^) was accomplished by intraperitoneal injection (IP) in four-week-old animals. We inoculated 10^5^ p.f.u. per animal. Following inoculation, the mice were visually monitored daily and scored for morbidity and mortality. For all the experiments, the mice were bred and kept during the inoculation experiments in a specific-pathogen-free facility at Instituto René Rachou. The mice were maintained in a temperature- and humidity-controlled facility on a 12 h light/dark cycle with food and water ad libitum. The AG129 mice were bred and maintained at the Animal Facility of the Instituto René Rachou, Fiocruz Minas. The experiments were approved by the Institutional Animal Care and Use Committee, Comissão de Ética no Uso de Animais da Fiocruz (CEUA) and performed according to institutional guidelines (license number LW-26-20).

### 2.4. Mosquito Infection with CHIKV

Five- to seven-day-old female mosquitoes were transferred into cylindrical containers fitted with nylon mesh (0.88 mm hole size) on top and starved through sugar deprivation for 24 h prior to mice blood feeding. All the mosquito infections were performed using AG129 mice. Infected AG129 mice were anesthetized two days post-infection using ketamine/xylazine (80/8 mg per kg^−1^). Subsequently, anesthetized mice were placed on the top of the netting-covered containers with female mosquitoes. Unshaved mice were placed in a prone position, with the entire ventral surface and limbs available to the mosquitoes. A maximum of 30 female mosquitoes were allowed to feed on 1 mouse for 30 min. After blood feeding, fully engorged females were selected. All the engorged females were placed in a container covered with nylon mesh with a cotton pad soaked with 10% glucose solution and with a plastic cup with soaked paper on the bottom for egg laying. Mice-fed female mosquitoes were harvested individually for tissue dissection and subsequent RNA extraction at four- and eight-days post-feeding.

### 2.5. CHIKV Transmission from Mosquitoes to AG129 Mice

To evaluate the mosquito populations’ transmission efficiency of CHIKV, we first fed five- to seven-day-old female mosquitoes on viremic AG129 mice (maximum of 30 female mosquitoes were allowed to feed on 1 mouse for 30 min). After blood feeding, fully engorged females were selected and placed in a container covered with nylon mesh with a cotton pad soaked with 10% glucose solution and with a plastic cup with soaked paper on the bottom for egg laying. Eight days after the infection, the female mosquitoes were allowed to feed on two- to three-week-old anesthetized naive AG129 mice. Each AG129 mouse was exposed to one female mosquito for a duration of 30 min, allowing for feeding. After blood feeding, fully engorged females were selected and harvested individually for RNA extraction for arbovirus quantification. Three days after the AG129 mice were exposed to the mosquitoes, blood was collected for arbovirus quantification by RT-qPCR.

### 2.6. RNA Extraction and RT-qPCR

RNA extraction from serum samples or mosquito samples was performed using the TRIzol reagent method (Invitrogen, Carlsbad, CA, USA), as previously described [38]. The total RNA was extracted, according to the manufacturer’s instructions, with some modifications. Briefly, 50 µL of serum or whole mosquito was placed in a 1.5 mL Eppendorf tube, and then 200 µL of TRIzol and two glass beads were added. Then, the samples were grounded vigorously in a bead beater for 90 s. Next, the samples were left for 10 min at room temperature and then 40 µL of chloroform was added to the tube and mixed vigorously for 30 s. After 10 min at room temperature, the samples were centrifuged at 12,000× *g* for 15 min at 4 °C. Then, the supernatant was mixed with an equal volume of chilled isopropanol by reversed mixing for 2 min and left overnight at −20 °C. Next, the samples were centrifuged at 12,000× *g* for 5 min at 4 °C. The sediment was washed with 75% (*v*/*v*) ethanol and air-dried for about 10 min. The purified RNA was dissolved in 20 µL of RNase-free water and stored at −80 °C. The total RNA extracted was reverse transcribed using M-MLV reverse transcriptase (Promega, Madison, WI, USA) and using random primers for initiation. Negative controls were prepared following the same protocol, without adding the reverse transcriptase. All the real-time PCR reactions were performed using the QuantStudio 12K Real-Time PCR System (Applied Biosystems, Foster City, CA, USA), and the amplifications were performed using the SYBR Green PCR Master Mix (Applied Biosystems—Life Technologies, Foster City, CA, USA). The final reaction volume was 10 µL. The thermal cycling conditions were composed of a Hold Stage (fast ramp to 95 °C, hold 20 s); PCR Stage (40 cycles of 95 °C, hold 15 s, fast ramp to 60 °C, hold 60 s); and Melt Stage (fast ramp to 95 °C, hold 15 s, fast ramp to 60 °C, hold 1 min, slow ramp of 0.05 °C/s to 95 °C, hold 15 s). All the real-time PCR reactions were carried out in triplicate. The relative quantification of gene expression was determined using the 2∆Ct method, as previously described [39]. Briefly, we employed the 2^−ΔCt^ (delta Ct) method, wherein the Ct (cycle threshold) values of the target gene (CHIKV gene) were normalized relative to the Ct values of the internal reference gene (housekeeping gene *RPL32*) within the same sample. The viral RNA load was expressed relative to the endogenous control housekeeping gene *RPL32* for *Ae. aegypti* and *RPL32* for AG129 mice. For *Ae. aegypti*, the *RPL32* primers were: Forward: 5′-AGC CGC GTG TTG TAC TCT G-3′ and Reverse: 5′-ACTTCT TCG TCC GCT TCT TG-3′. For mice, the *RPL32* primers were: Forward: 5′-GCTGCC ATC TGT TTT ACG G-3′ and Reverse: 5′-TGA CTG GTG CCT GAT GAA CT-3′. For CHIKV, the primers were: Forward: 5′-AAG CTY CGC GTC CTT TAC CAA G-3′ and Reverse: 50-CCA AATTGT CCY GGT CTT CCT-3′.

### 2.7. Statistical Analyses

The statistical analyses were conducted using the software R v4.2.3 (www.r-project.org) assessed on 10 April 2023. Logistic regression analyses were performed to assess the association between the predictor variable (the different populations of *Ae. aegypti*) and the binary outcome variable (infection status) using a Generalized Linear Model (GLM) with the family argument set to “binomial” and the link function to “logit” followed likelihood-ratio χ^2^ tests. To compare the differences in viral loads in the midgut and carcass samples among the mosquito populations, we performed a Kruskal–Wallis test followed by a post hoc test (Dunn’s test) to determine the pairwise differences between the groups. *p*-values less than 0.05 were considered statistically significant [37].

## 3. Results

To assess the vector competence of Brazilian *Ae. aegypti* mosquito populations for CHIKV, we analyzed the susceptibility to infection, the ability of the virus to disseminate out of the midgut and the potential of being transmitted to a vertebrate host. For that, we used a CHIKV-ECSA strain recently isolated in Brazil. These analyses were performed for four different *Ae. aegypti* populations covering four different Brazil states: Araraquara city in São Paulo state (ARA), Jaboticatubas city in Minas Gerais state (JAB), Petrolina state in Pernambuco state (PET) and Porto Alegre city in Rio Grande do Sul state (POA) (Figure 1 and Table 1). For that, we exposed five- to seven-day-old female mosquitoes from all four populations to the same CHIKV-infected AG129 mice for blood-feeding for 30 min. At four- and eight-days post-feeding (d.p.f.), we collected the mosquitoes and analyzed the presence of CHIKV in the midgut and carcass to evaluate the infection rates and dissemination efficiency, respectively. Additionally, at eight days post-feeding, another set of mosquitoes was individually exposed to naïve AG129 mice to evaluate the transmission efficiency.

### 3.1. Different Populations of Aedes aegypti across Brazil Exhibit High and Similar CHKIV Infection Rates as Well as Dissemination Efficiency

Using the locally isolated CHIKV-ECSA strain, we assessed the infection rates, which were calculated as the proportion of blood-fed mosquitoes that tested positive for the virus in the midgut relative to the total number of blood-fed mosquitoes, in four *Ae. aegypti* populations. We found that the infection rates were consistently high and uniform across all four Brazilian populations tested, starting as early as 4 days post-feeding and continuing through to 8 days post-feeding (Figure 2B,D), ranging from 82% at 8 d.p.f for the Porto Alegre population (POA) to 100% at 4 d.p.f. for the Jaboticatubas population (JAB). Using a logistic regression model to estimate the relationship between the infection rates and the different populations, we found no significant differences between the four mosquito populations for both time points: 4 d.p.f (*p* = 0.2093, Appendix A) and 8 d.p.f. (*p* = 0.5197, Appendix A). Even when we analyzed the data from both time points together, we found no significant differences between the populations (*p* = 0.7393). As shown in Figure 2C, the midgut viral loads at 4 d.p.f. were similar between the four mosquito populations tested (Kruskal–Wallis test). However, at 8 d.p.f., we observed that the Petrolina population (PET) exhibited significantly higher amounts of CHIKV in the midgut compared to the other three populations tested (PET–ARA *p =* 0.00465; PET–JAB *p =* 0.000156; PET–POA *p* < 0.0001, Kruskal–Wallis test, Figure 2E).

### 3.2. CHIKV Shows High Dissemination Efficiency in Mosquitoes of Brazilian Aedes aegypti Populations

To compare how well CHIKV spreads from the mosquito midgut to other body parts in the different *Ae. aegypti* populations, we measured the percentage of blood-fed mosquitoes with the virus in their carcass (their bodies, excluding the midgut) out of the total number of blood-fed mosquitoes from four Brazilian populations. Notably, we observed that CHIKV was already present in the carcass as early as 4 d.p.f. in most of the mosquitoes tested (Figure 3A). For both the 4- and 8-day d.p.f. time points, we observed consistently high dissemination rates across all four populations tested, ranging from 95% for the Araraquara population at 4 days post-feeding to 80% for the Porto Alegre mosquitoes at 8 days post-feeding (Figure 3A,C). Nevertheless, no significant differences were found between the populations for either the 4- or 8-day d.p.f. time points (*p* = 0.012 and *p* = 0.6602, respectively; see Appendix A). We observed some variation in the viral load in the carcasses of the different mosquito populations (Figure 3B); however, this variation was not significantly different for most comparisons, except between Petrolina and Porto Alegre (*p* = 0.01, Kruskal–Wallis test). It is interesting to note that this variation appears to have disappeared by 8 days post-infection (Figure 3D).

### 3.3. High Transmission Efficiency of CHIKV across Brazilian Aedes aegypti Populations

To assess the transmission efficiency of the CHIKV-ECSA strain, we exposed female mosquitoes, which had an infectious blood meal 8 days previously, to AG129 naive mice for a second blood meal. After exposing each female to one different AG129 mouse, we were able to observe that the majority of the mice become infected with CHIKV (Figure 4B). As shown in Figure 4B, we found that the transmission efficiency (the percentage of blood-fed mosquitoes that were able to infect the mouse out of the total number of blood-fed mosquitoes) varied from 80% in the Araraquara (ARA) mosquitoes to 67% in the Petrolina (PET) mosquitoes. Despite the observed variations, when estimating the relationship between the transmission efficiencies and the different populations using a logistic regression model, we found no significant differences between the four mosquito populations (*p* = 0.9302, Appendix A).

## 4. Discussion

CHIKV has emerged as a global health threat over the past three decades, expanding from tropical and subtropical regions of Africa to Indian Ocean islands, the Indian subcontinent, South Asia, South America, and some regions of Europe [2,7]. Although two CHIKV lineages are confirmed to be circulating in Brazil, the Asian lineage and the ECSA lineage, surveillance studies indicate that the ECSA lineage is currently the prevailing genotype circulation in the country [25,26,27,28]. In this regard, evaluating the *Ae. aegypti* vector competence for the currently circulating CHIKV strain in this region is crucial to gain a better understanding of the role of the *Ae. aegypti* vector in CHIKV epidemiology in Brazil. To address this, we evaluated the infection rates, dissemination, and transmission efficiencies of an ECSA strain isolated in 2018 from Niterói, Brazil, across four *Ae. aegypti* populations collected from different regions of the country.

Using the viremic Ag129 mice to infect the mosquitoes, our results confirm that the CHIKV-ECSA lineage exhibits a high level of infectivity across all the Brazilian *Ae. aegypti* populations that were tested in this study, with infection rates reaching as high as 100% and no lower than 80% (Figure 2B,D). Notably, the results also suggest that the CHIKV-ECSA lineage is able to disseminate out of the midgut at a very early stage, since at just 4 days after the infectious blood meal, the majority of mosquitoes tested positive for the virus in their carcasses (Figure 3A,B). However, it is important to note that we cannot dismiss the possibility that some virus remained in the proboscis of the mosquitoes at this time point, considering that the carcasses included the head. Our results confirm previous findings that *Ae. aegypti* populations in South America exhibit high CHIKV dissemination efficiencies [29]. However, to the best of our knowledge, this study is the first to report high infection rates and dissemination efficiencies using a locally isolated CHIKV strain that represents the currently circulating genotypes. Furthermore, by using Ag129 mice to evaluate the transmission competence, which better resembles the natural cycle of transmission in nature, we have shown that the Brazilian populations are capable of efficiently transmitting the CHIKV-ECSA lineage, with transmission efficiencies ranging from 67% to 80%.

Overall, our results indicate that local *Ae. aegypti* populations are highly competent in being infected with and transmitting the CHIKV-ECSA strain circulating in Brazil. This suggests that Brazilian *Ae. aegypti* mosquitoes could play a significant role in disseminating this lineage across the country. These observations may help to explain why the CHIKV-ECSA lineage has experienced a significant expansion in Brazil in recent years, as well as the explosive emergence of CHIKV in Brazil and other South American countries. However, it is essential to note that the CHIKV Asian lineage is also present in Brazilian territories, making it important to evaluate the susceptibility of *Ae. aegypti* populations to the local genotypes of this lineage. Additionally, other factors may also play a role in the epidemiology of CHIKV in Brazil. For example, the high population densities in urban areas and the higher prevalence of *Ae. aegypti* over *Ae. albopictus* in these areas could also contribute to the transmission of the virus. Although *Ae. albopictus* has a more peri-urban distribution in Brazil, previous studies have reported that this species is a competent vector for different lineages of CHIKV. Therefore, it is also crucial to assess the vector competence of local populations of this species for currently circulating CHIKV lineages in the South American regions. Taken together, our results may contribute to explaining the recent epidemiology of CHIKV in Brazil and the increasing number of outbreaks associated with this ECSA lineage. Nonetheless, to gain a better understanding of the epidemiological dynamics of CHIKV, comprehensive studies are crucial, comparing currently circulating genotypes and including populations of both *Ae. aegypti* and Ae. *albopictus*. This information is vital for the development and implementation of effective strategies to control the spread of CHIKV and reduce the burden of this arbovirus on human populations.

## Figures and Tables

**Figure 1 viruses-16-00575-f001:**
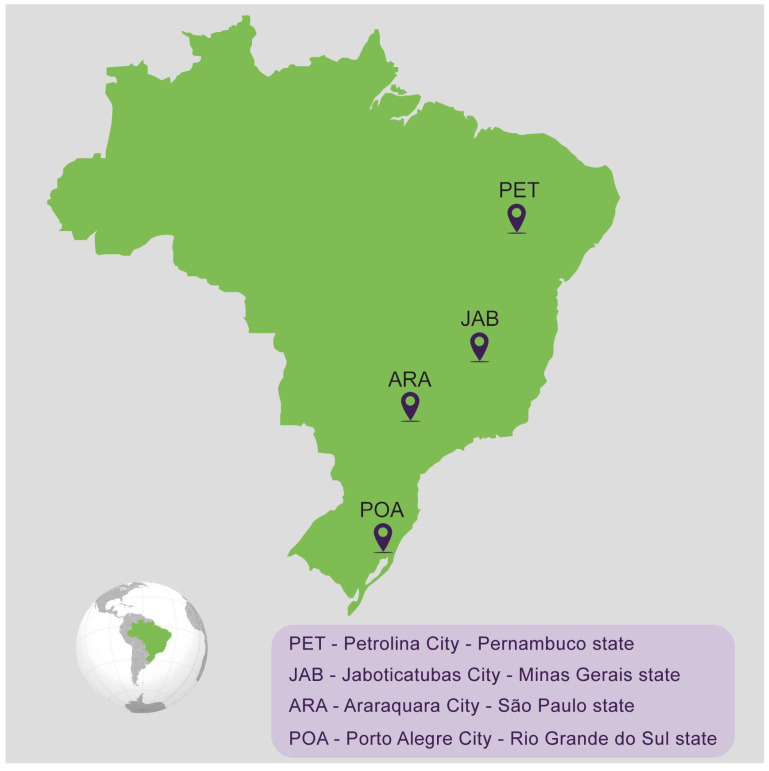
Map of Brazil showing the *Aedes aegypti* sampling sites.

**Figure 2 viruses-16-00575-f002:**
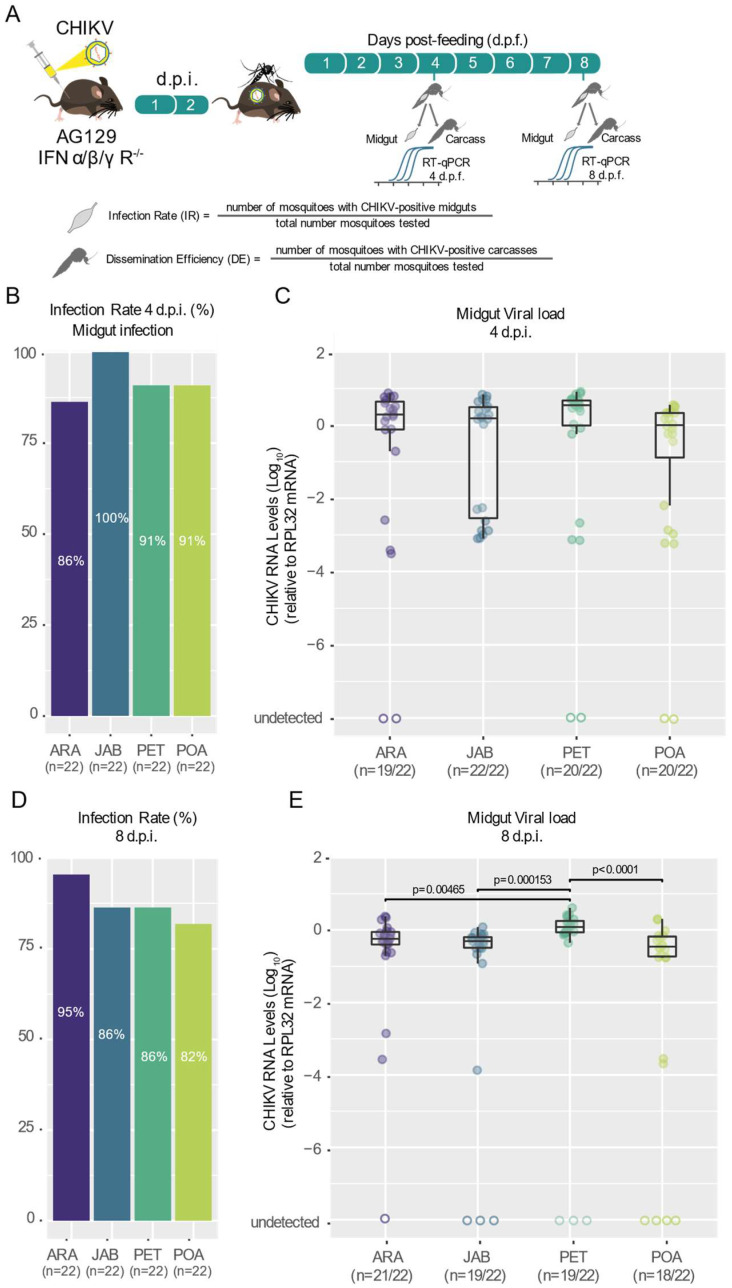
High infection rates of CHIKV across Brazilian *Aedes aegypti* populations. (**A**) Scheme of the experimental design used. Infection rates at 4 (**B**) and 8 (**D**) days post-feeding (d.p.f.) are shown. The population origin is indicated below each bar plot (ARA, JAB, PET, POA). The number of mosquitoes tested is indicated below the population abbreviation. The CHIKV RNA levels of each mosquito tested are shown. Significant differences are indicated by the *p*-value (Kruskal–Wallis test). (**C**,**E**) The number of mosquitoes with detected CHIKV RNA out of the total number of mosquitoes tested is presented below the population abbreviation. To quantify the CHIKV load, we utilized the 2^−ΔCt^ (delta Ct) method. The limit of detection was 10^10^. The mosquitoes used to obtain the results in both the midgut and the carcass were the same individuals from the cohort. Empty circles represent samples where viral RNA was not detected.

**Figure 3 viruses-16-00575-f003:**
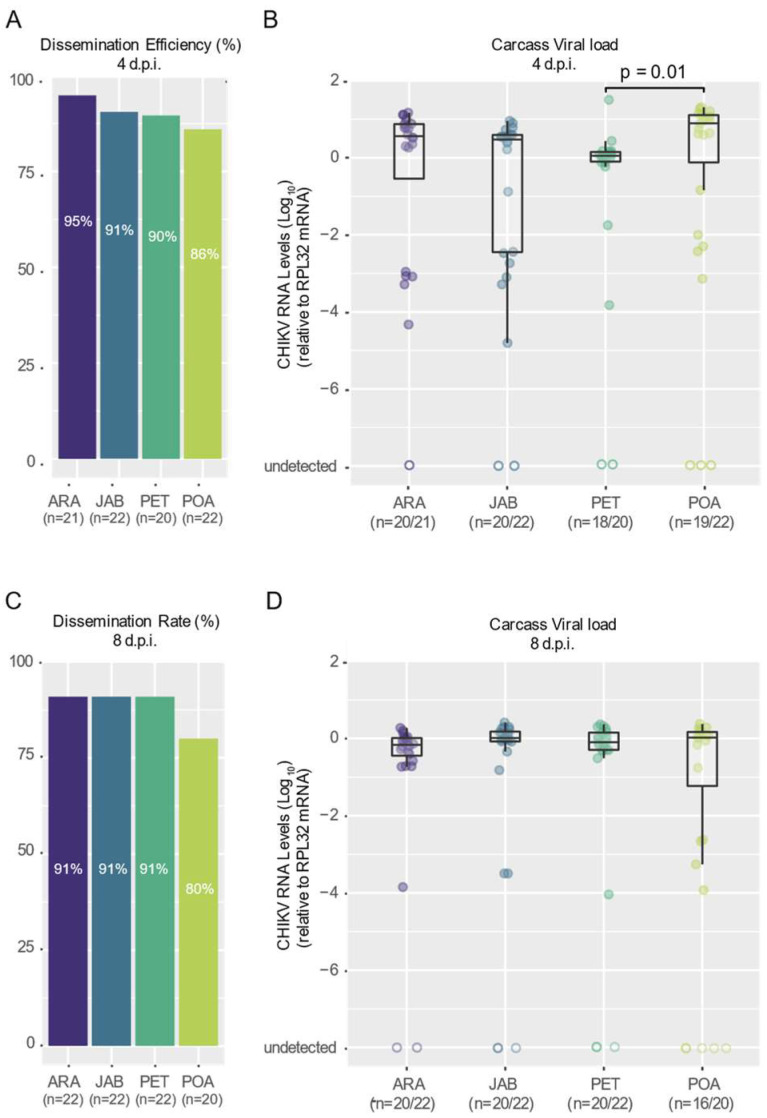
High dissemination efficiency of CHIKV across Brazilian *Aedes aegypti* populations. An ECSA strain of CHIKV isolated from a human patient in Niterói, Brazil, in 2018 was used. Dissemination efficiencies at 4 (**A**) and 8 (**C**) days post-feeding (d.p.f.) are shown. The population origin is indicated below each bar plot (ARA, JAB, PET, POA). The number of mosquitoes tested is indicated below the population abbreviation. The CHIKV RNA levels of each mosquito tested at 4 (**B**) and 8 (**D**) days post-feeding (d.p.f.) are shown. Significant differences are indicated by the *p*-value (Kruskal–Wallis test). The number of mosquitoes with detected CHIKV RNA out of the total number of mosquitoes tested is presented below the population abbreviation. Empty circles represent samples where viral RNA was not detected.

**Figure 4 viruses-16-00575-f004:**
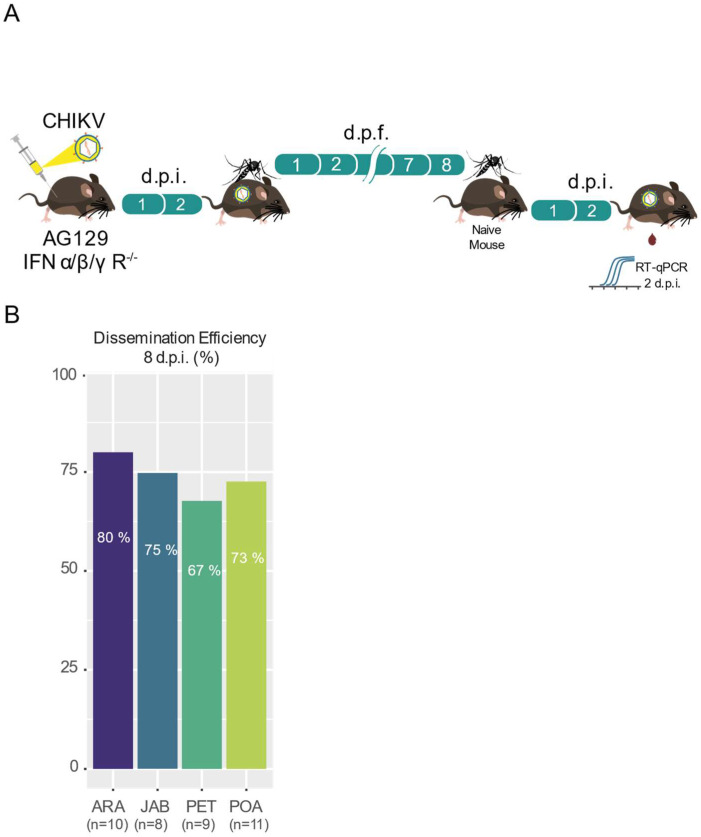
Brazilian *Aedes aegypti* populations are highly efficient in transmitting CHIKV. (**A**) Scheme of the experimental design used to assess the CHIKV transmission efficiency in *Aedes aegypti*. (**B**) Transmission efficiency at 8 days post-feeding (d.p.f.). The population origin is indicated below each bar plot (ARA, JAB, PET, POA). The number of mosquitoes tested is indicated below the population abbreviation.

**Table 1 viruses-16-00575-t001:** *Aedes aegypti* populations collected in Brazil.

Population Name	City	State	Species	Stage Collected	Number of Mosquitoes Used to Establish the Laboratory Population	Generation Used in the Experiments	Year of Collection
ARA	Araraquara	São Paulo	*Aedes aegypti*	eggs	100 females + 100 males	F2	2022
JAB	Jaboticatubas	Minas Gerais	*Aedes aegypti*	eggs	100 females + 100 males	F2	2022
PET	Petrolina	Pernambuco	*Aedes aegypti*	eggs	100 females + 100 males	F3	2022
POA	Porto Alegre	Rio Grande do Sul	*Aedes aegypti*	eggs	100 females + 100 males	F3	2022

## Data Availability

The data presented in this study are openly available in FigShare at https://doi.org/10.6084/m9.figshare.24415657.v1 (accessed on 29 March 2024).

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
