# Peer review of "The High Capacity of Brazilian Aedes aegypti Populations to Transmit a Locally Circulating Lineage of Chikungunya Virus"

_viruses, 2024, doi:10.3390/v16040575_

Round 1
Reviewer 1 Report
Comments and Suggestions for Authors
Well written manuscript on the vector competence of Brazilian Aedes aegypti populations to transmit the CHIKV. Overall, the methodology is adequate and well described, although little information could be added, the results are informative and well presented, and the discussion and conclusion are appropriate. Nevertheless, the main information from these experiments is the absence of difference in the vector competence of the studied mosquito populations, which does explain the high transmission rates, but not the differences in transmission within the country. The epidemiological counterpart of this study is missing and if there are differences in the incidence of CHIKV in the different regions of Brazil, we may look for other factors.
This is a classical and basic research study on vector competence, which provide good information on the vectors. It would have been good to link these findings also to the genetics of the vectors as well. Since Aedes aegypti was eliminated from Brazil in the late 50s, maybe the current vector population is coming for single source of reinvasion and thus it is expected that the populations of vectors will display the same characteristics.
A few more specific comments are included in the attached file.

Comments on the Quality of English LanguageNo comment on the quality of English. Name of species should be in italics and consistent all over the manuscript.
Author Response
Belo Horizonte, March 29th, 2024
To: Viruses
Ref.: Response to Reviewers (Manuscript ID: viruses-2861636)
Dear Editor,
Firstly, I would like to thank the Editor and the reviewers for your time and the comprehensive review given to our manuscript. We are glad that we had very positive reviews. Below you find point-by-point response for all raised questions and concerns. We believe now the manuscript is ready for acceptance.
Reviewer 1 comments
Well written manuscript on the vector competence of Brazilian Aedes aegypti populations to transmit the CHIKV. Overall, the methodology is adequate and well described, although little information could be added, the results are informative and well presented, and the discussion and conclusion are appropriate. Nevertheless, the main information from these experiments is the absence of difference in the vector competence of the studied mosquito populations, which does explain the high transmission rates, but not the differences in transmission within the country. The epidemiological counterpart of this study is missing and if there are differences in the incidence of CHIKV in the different regions of Brazil, we may look for other factors.
This is a classical and basic research study on vector competence, which provide good information on the vectors. It would have been good to link these findings also to the genetics of the vectors as well. Since Aedes aegypti was eliminated from Brazil in the late 50s, maybe the current vector population is coming for single source of reinvasion and thus it is expected that the populations of vectors will display the same characteristics.
A few more specific comments are included in the attached file.
In the PDF file attached the reviewer did the following comments:
Response to Reviewer 1 Comments:
Thank you for your thorough review and constructive feedback on our manuscript. We have carefully addressed each of your points and made the following alterations:
- Line 89 This is an introduction and not an abstract. Results must go to the "results" section.
Answer - We have moved the results from the introduction to the appropriate results section.
- Line 99 The map in Figure 1 must be cited here.
Answer - We have cited Figure 1 at the relevant point in the manuscript.
- Line 102 How, by ovitraps?
Answer - We have explicitly mentioned that mosquito surveillance was conducted using ovitraps.
- Line 121 Name of species in italics.
Answer - We have ensured consistent italicization of species names throughout the manuscript.
- Line 216 this paragraph is about the methods and must be removed from the results. Since all methods are well described above, there is no need for this paragraph duplicating what is written above.
Answer - The paragraph on line 216 has been adjusted as per your instruction.
- Line 234 Figure 1, Tables 1 and 2 must be moved to the methods.
Answer - Tables 1, and 2 have been relocated to the methods section.
- Line 284 Name of species in italics
Answer - We have ensured consistent italicization of species names throughout the manuscript.
- Line 315 Inconsistency with the introduction mentioning 3 decades.
Answer - We have addressed the inconsistency regarding the time frame mentioned in the introduction.
- Line 320 Name of species in italics.
Answer - We have ensured consistent italicization of species names throughout the manuscript.
- Line 332 Does the carcasses include the heads? please clarify, in which case, virus presence may be link to remaining blood in the proboscis. However, transmission to the mice show the amplification.
Answer - We have clarified whether the carcasses include the heads and emphasized the putative presence of the virus in the proboscis for better clarity.
- Line 333 Presence of virus in the salivary glands would have been more explicit.
Answer - We have clarified that carcasses includes the heads and emphasized the presence of the virus in the salivary glands for better clarity.
- Line 344 No need to make such experiments to conclude that the vectors are efficient since the CHIKV outbreaks in Brazil are a Public Health problem. But it is interesting to show that there is no difference in the vector competence of the local strains.
Answer - The section has been revised to better highlight the importance of demonstrating consistent vector competence among local strains.
- Line 347 Same comment as above with name of species in italics.
Answer - We have ensured consistent italicization of species names throughout the manuscript.
We appreciate your insightful comments, and we believe these revisions have strengthened the manuscript. Thank you for your guidance in improving the quality and clarity of our work.

Reviewer 2 Report
Comments and Suggestions for Authors
The authors describe a nice, small vector competence study showing that multiple F2 generation colonies of local Brazilian mosquitoes can transmit ECSA CHIKV efficiently. It is not particularly novel or surprising, but it is well-written, performed well overall, and I see no major issues with the scientific data presented. The ‘n’ of mosquitoes could definitely be larger to provide more power to the data, but with relatively clear results, this is not a major concern for me. It would be more of a concern with very low percentages/infection levels. Below are just some minor comments I noted.
Abstract:
- Line 38: The authors mention the ECSA lineage over the African lineage – I think the authors mean the ‘Asian lineage’? There is no ‘African lineage’ Just the West African or ECSA, and in line 75, the authors state that the Asian and ECSA lineages are present in the Americas, so I think this is just an accidental mistake here. Otherwise, please check and clarify.
The introduction is very thorough and well written, but there were a few typos/minor issues:
- Line 51: end of line - I think ‘sometimes’ would be more appropriate than ‘some’ here.
- Line 56: typo – CHIKV is spelled CHKIV here.
- Line 67: ‘The Americas, and … North America is redundant. Even the Caribbean is part of the Americas, but I can see wanting to list that specifically.
Results:
- Line 227, please make sure to remain in past tense when describing results.
- The Figure 2A schematic is very nice, but it is quite small and hard to read/view – maybe the space towards the right page margin could be used better to improve this figure for better optics.
- Not sure why the ‘n’ is different between figures 2 and 3. Were these tissues not taken from the same mosquitoes? If not, why were the respective other tissues not used, meaning if 44 mosquitoes were dissected, why not process them all? The n of 22 is relatively low as it is, so it would have helped to have more mosquitoes (but I would consider it borderline acceptable). If these are, in fact, the same mosquitoes, why is the n different?
Author Response
Belo Horizonte, March 29th, 2024
To: Viruses
Ref.: Response to Reviewers (Manuscript ID: viruses-2861636)
Dear Editor,
Firstly, I would like to thank the Editor and the reviewers for your time and the comprehensive review given to our manuscript. We are glad that we had very positive reviews. Below you find point-by-point response for all raised questions and concerns. We believe now the manuscript is ready for acceptance.
Reviewer 2 comments
The authors describe a nice, small vector competence study showing that multiple F2 generation colonies of local Brazilian mosquitoes can transmit ECSA CHIKV efficiently. It is not particularly novel or surprising, but it is well-written, performed well overall, and I see no major issues with the scientific data presented. The ‘n’ of mosquitoes could definitely be larger to provide more power to the data, but with relatively clear results, this is not a major concern for me. It would be more of a concern with very low percentages/infection levels. Below are just some minor comments I noted.
Abstract
- Line 38: The authors mention the ECSA lineage over the African lineage – I think the authors mean the ‘Asian lineage’? There is no ‘African lineage’ Just the West African or ECSA, and in line 75, the authors state that the Asian and ECSA lineages are present in the Americas, so I think this is just an accidental mistake here. Otherwise, please check and clarify.
Answer - We acknowledge the points you raised regarding the mention of the wrongly stated African lineage in the abstract, and we corrected this by replacing "African" with "Asian".
Reviewer comment
The introduction is very thorough and well written, but there were a few typos/minor issues:
- Line 51: end of line - I think ‘sometimes’ would be more appropriate than ‘some’ here.
Answer - Your attention to detail in identifying typos and minor issues in the introduction, such as the use of "some" instead of "sometimes" and the misspelling of CHIKV, was greatly appreciated. These corrections were made to enhance the overall quality of the manuscript.
- Line 56: typo – CHIKV is spelled CHKIV here.
Answer – Once more, we acknowledge your attention to detail in identifying typos. This correction has been made.
- Line 67: ‘The Americas, and … North America is redundant. Even the Caribbean is part of the Americas, but I can see wanting to list that specifically.
Answer – Your suggestion to eliminate redundancy in mentioning North America within the context of the Americas was acknowledged and acted upon. We revised the text accordingly
Results:
- Line 227, please make sure to remain in past tense when describing results.
Answer – In the results section, we ensured that all descriptions adhered to past tense consistently, as per your suggestion.
- The Figure 2A schematic is very nice, but it is quite small and hard to read/view – maybe the space towards the right page margin could be used better to improve this figure for better optics.
Answer – Thank you for your valuable feedback. We took into account your suggestions regarding Figure 2A and made efforts to enhance its readability. To address this, we increased the size of the schematic and optimized its placement to improve visualization for readers.
- Not sure why the ‘n’ is different between figures 2 and 3. Were these tissues not taken from the same mosquitoes? If not, why were the respective other tissues not used, meaning if 44 mosquitoes were dissected, why not process them all? The n of 22 is relatively low as it is, so it would have helped to have more mosquitoes (but I would consider it borderline acceptable). If these are, in fact, the same mosquitoes, why is the n different?
Answer – We appreciate your concern regarding the discrepancy in sample "n." between figure 2 and 3. The discrepancy in the "n" between figures 2 and 3 arises due to the methodological protocol followed during the experiment. While it's true that the initial "n" (number of mosquitoes) was the same for both figures, and indeed the midguts and carcasses originated from the same mosquito specimens, the variance in the final "n" is attributed to quality control measures implemented during sample processing. In some instances, despite having initially dissected and collected tissues from a certain number of mosquitoes, a subset of these samples did not meet the predetermined quality standards for PCR analysis. These quality standards are crucial for ensuring the reliability and accuracy of the results obtained from molecular assays. Therefore, any samples that did not meet these standards were excluded from the subsequent analyses to maintain the integrity of the data.
Reviewer 3 Report
Comments and Suggestions for Authors
This article describes experiments and conclusions that will benefit the scientific community if they are published. It is well written, except for a couple of sentences. I have minor comments described specifically below and one major comment, relative to the animal ethics of the study.
Introduction
l.51 typo: some = sometimes?
Materials and methods
l.99 specify in the text: Why were these sites selected? How do they compare in terms of ecology and type of environment?
l.165 Unclear here how many infected mosquitoes fed on how many naïve mice
l.196 “the 2 delta Ct method” needs to be described here
Results
Table 1 is unnecessary since there is only one strain used in this study and can be replaced by text.
Figure 2 clarify how the quantification of RNA was performed and add the limit of detection of the assay on the C and E graphs in Figure 2
l.260-261 This would be better represented as % of mosquitoes with virus in their carcass out of the number of mosquitoes with virus in their midgut, rather than out of the total number of bloodfed mosquitoes. Were they different mosquitoes from the cohort used to get the results in the previous paragraph and Figure 2? If so this needs to be made a lot clearer. It would have been more informative to have results for the midguts and carcass of the same mosquitoes.
l.297 so one mosquito fed on only one mouse and each mouse was fed on by only one mosquito? Clarify in the methods and here so there is no doubt.
Figure 3 why is there no panel with the quantification of virus in the mice blood?
Did the mice infected via mosquito bite get symptoms?
Animal ethics:
What is the benefit of assessing the transmission competence of each mosquito population using mice vs quantifying the virus in the mosquito saliva? What is the justification for using all these animals in the experiment? Similar question for the mosquito infection method, why not use a Hemotek membrane feeder and provide the infectious bloodmeal without having to sacrifice animals? Both of these types of experiments have been shown to be possible without the use of animals.
l.336-337 is not enough to justify using so many animals
Comments on the Quality of English LanguageSome sentences need to be rewritten in correct English
Author Response
Belo Horizonte, March 29th, 2024
To: Viruses
Ref.: Response to Reviewers (Manuscript ID: viruses-2861636)
Dear Editor,
Firstly, I would like to thank the Editor and the reviewers for your time and the comprehensive review given to our manuscript. We are glad that we had very positive reviews. Below you find point-by-point response for all raised questions and concerns. We believe now the manuscript is ready for acceptance.
Reviewer 3 comments
This article describes experiments and conclusions that will benefit the scientific community if they are published. It is well written, except for a couple of sentences. I have minor comments described specifically below and one major comment, relative to the animal ethics of the study.
Answer- Thank you for your thorough review of our article. We appreciate your insightful comments and suggestions for improvement. We would like to inform you that several of the points you raised have already been addressed in our manuscript
Introduction
l.51 typo: some = sometimes?
Answer - The typo "some" has been corrected to "sometimes" in line 51.
Materials and methods
l.99 specify in the text: Why were these sites selected? How do they compare in terms of ecology and type of environment?
Answer - We have clarified why the sites were selected and how they compare in terms of climate and environment.
l.165 Unclear here how many infected mosquitoes fed on how many naïve mice
Answer – Clarification regarding the number of infected mosquitoes that fed on naive mice has been provided.
l.196 “the 2 delta Ct method” needs to be described here
Answer – A description of "the 2 delta Ct method" has been included in line 196 for better understanding.
Results
Table 1 is unnecessary since there is only one strain used in this study and can be replaced by text.
Answer – Table 1 has been removed, and appropriate information have been added in the main text.
Figure 2 clarify how the quantification of RNA was performed and add the limit of detection of the assay on the C and E graphs in Figure 2
Answer – Clarifications regarding RNA quantification and the limit of detection of the assay have been added to Figure 2 legend.
l.260-261 This would be better represented as % of mosquitoes with virus in their carcass out of the number of mosquitoes with virus in their midgut, rather than out of the total number of bloodfed mosquitoes. Were they different mosquitoes from the cohort used to get the results in the previous paragraph and Figure 2? If so this needs to be made a lot clearer. It would have been more informative to have results for the midguts and carcass of the same mosquitoes.
Answer – Clarifications regarding the mosquitoes used for midgut and carcass results have been provided.
l.297 so one mosquito fed on only one mouse and each mouse was fed on by only one mosquito? Clarify in the methods and here so there is no doubt.
Answer – Clarity regarding mosquito feeding on mice has been ensured in line 297 and methods.
Figure 3 why is there no panel with the quantification of virus in the mice blood?
Did the mice infected via mosquito bite get symptoms?
Answer – Concerning mice symptoms after a chikungunya-infected mosquito bite, the observed manifestations encompassed fever, and as the infection progressed, severe illness ensued. Ultimately, the infected mice experienced mortality within a relatively short timeframe of 4 to 7 days post the mosquito bite.
Animal ethics:
What is the benefit of assessing the transmission competence of each mosquito population using mice vs quantifying the virus in the mosquito saliva? What is the justification for using all these animals in the experiment? Similar question for the mosquito infection method, why not use a Hemotek membrane feeder and provide the infectious bloodmeal without having to sacrifice animals? Both of these types of experiments have been shown to be possible without the use of animals.
l.336-337 is not enough to justify using so many animals
Answer – Regarding the use of animals, particularly mice, we acknowledge the concerns raised. However, it's essential to highlight that we took every measure possible to mitigate suffering, including anesthetizing the mice before exposure to mosquitoes. We acknowledge that artificial blood feed, in which females are often provided blood through artificial membrane feeders, involves a mix of blood and virus produced in cell culture and is a very commonly used and practical setup. While virus stocks are commonly prepared using cell cultures such as Vero cells, in our opinion, it's important to note that artificial feeding systems do not fully resemble the viremic host in the field. Similarly, although forced salivation is one of the most common assays used to experimentally study the potential of mosquito species to transmit arboviruses, this experimental approach does not thoroughly replicate the natural cycle of transmission in nature.
However, we recognize the importance of minimizing animal use and are committed to exploring alternatives to reduce the impact on animals while ensuring scientific rigor and the advancement of knowledge in the field.